# Early Experience with Inner Branch Stent–Graft System for Endovascular Repair of Thoraco-Abdominal and Pararenal Abdominal Aortic Aneurysm

**DOI:** 10.3390/diagnostics14232612

**Published:** 2024-11-21

**Authors:** Simone Cuozzo, Antonio Marzano, Ombretta Martinelli, Jihad Jabbour, Andrea Molinari, Vincenzo Brizzi, Enrico Sbarigia

**Affiliations:** 1Vascular Surgery Division, Department of Surgery “Paride Stefanini”, Policlinico Umberto I “La Sapienza” University of Rome, 00161 Rome, Italy; simone.cuozzo89@gmail.com (S.C.); ombretta.martinelli@uniroma1.it (O.M.); drjihadjabbour@gmail.com (J.J.); andrea.molinari@uniroma1.it (A.M.); enrico.sbarigia@uniroma1.it (E.S.); 2Vascular Surgery Department, CHU de Bordeaux, 33000 Bordeaux, France; vinbrizzi@hotmail.com

**Keywords:** thoraco-abdominal aortic aneurysm (TAAA), pararenal abdominal aortic aneurysm (PAAA), branched stent graft, E-nside multibranch stent–graft system, inner branch

## Abstract

Objectives: This study aims to evaluate the technical and clinical outcomes of the E-nside stent graft for thoraco-abdominal aortic aneurysm (TAAA) and pararenal abdominal aortic aneurysm (PAAA) endovascular treatment at our University Hospital Center. Methods: We conducted a retrospective analysis of patients electively treated by inner branched EVAR (iBEVAR) between 05/2021 and 03/2023. Demographic, procedural, and clinical data were analyzed. The technical success and clinical outcomes, such as access-site-related complications were reported. The perioperative and early mortality rate, freedom from aortic reintervention, target vessels’ (TVs) patency, and the endoleak rate were evaluated during the follow-up. The technical aspects (external iliac artery diameter, iliac tortuosity, extent of aortic coverage) were retrospectively analyzed. Results: Twenty-two patients were included (age 75.9 ± 5.5; 72.7% male). The aneurism extent was Crawford I = 4 (18.2%), III = 8 (36.4%), IV = 5 (22.7%), V = 1 (4.5%), and PAAA = 4 (18.2%). The mean aortic diameter was 63.5 ± 9.9 mm. The technical success was 95.5% (assisted primary success 100%). The clinical success was 86.4%. The perioperative and early freedom from all-cause mortality rates were 90.9% and 90%, respectively. No case of inter-stage aortic-related mortality was reported, and there was no permanent/temporary spinal cord ischemia (SCI). Seventy-eight out of 81 patent TVs were incorporated through a bridging stent (96.3%). The TV success was 95.1%. The mean external iliac artery (EIA) diameter was 7.5 ± 1.1 mm. Twelve patients (54.5%, including all female patients) were considered outside the instructions for use (IFU) due to narrow iliac arteries. One access-site-related complication was reported. Conclusions: Our experience confirms that E-nside has promising technical and clinical success rates, as well as a low reintervention rate, but it requires a significant compromise of the healthy aortic tissue and adequate iliac arteries that still represents a limitation, especially among women. Mid- to long-term studies and prospective registries are mandatory to evaluate the long-term efficacy and safety, as a comparison between E-nside and other alternative off-the-shelf solutions.

## 1. Introduction

The availability of multibranch stent–graft devices for endovascular repair of thoraco-abdominal aortic aneurysms (TAAA) represents a significant advancement in endovascular techniques. These devices offer a minimally invasive alternative to open surgery, potentially reducing the recovery time and minimizing the complications typically associated with more invasive procedures.

However, an endovascular approach is not suitable for every patient or all types of TAAAs. Therefore, detailed evaluation of the aortic and iliac axes’ anatomy and the patient’s selection by a multidisciplinary team are crucial for determining the most appropriate treatment option for each individual case [1]. It is well known that aorto-iliac morphology poses significant limitations impacting the feasibility of treating TAAAs with currently available stent grafts. The E-nside thoraco-abdominal multibranch stent–graft system (Artivion Inc., Kennesaw, GA, USA) is the first available precannulated, inner branch based, and off-the-shelf solution for endovascular TAAA repair. The inner branch technology enables the treatment of a wider range of aortic anatomies with a consistent approach: internal tunnels can be used in narrow and kinked anatomies as well as large aneurysms. Additionally, precannulation aims to reduce fluoroscopy and implantation time and to limit the use of contrast media.

This study aims to report the clinical and technical outcomes of our initial experience with the E-nside stent–graft system.

## 2. Materials and Methods

Patients who underwent endovascular repair of TAAA or pararenal abdominal aortic aneurysm (PAAA) with the E-nside stent–graft system between May 2021 and March 2023 were included in this retrospective single-center cohort study.

Clinical, technical, and follow-up data were collected, entered into a specifically maintained database, and retrospectively analyzed. Demographic characteristics, anatomical features, procedure-related complications, reinterventions, Intensive Care Unit (ICU) and in-hospital length of stay, and aneurysm- and all-causes-related mortality were recorded.

Preoperative computed tomography angiography (CTA) was used to classify aneurysms according to the Crawford classification.

### 2.1. Device Description

The E-nside device is designed as a tube graft made of a polytetrafluoroethylene (PTFE) fabric with a nitinol frame (Figure 1). A comprehensive technical description of the device has been previously published [2,3,4]; nonetheless, a brief summary is provided here. Proximally, the graft is designed to dock into a thoracic stent graft. Therefore, it has no ancillary proximal fixation (barbs), and it only relies on the outward radial force of the nitinol frame [4].

The stent graft is available in four configurations, with proximal diameters of 38 or 33 mm, a total length of 222 mm, and distal diameters of 30 or 26 mm. The central section has a diameter of 24 mm and includes four internal branches: 20 mm in length and 8 mm in diameter for the celiac trunk (CT) and superior mesenteric artery (SMA) and 20 mm in length and 6 mm in diameter for the renal arteries (Figure 1). The inner branches are precannulated with polyamide hypotubes to facilitate snaring and direct anterograde catheterization during the procedure. The device is loaded into a 24 F (outer diameter, OD) large bore sheath.

### 2.2. Procedural Details

All procedures were performed under general anesthesia by a team of vascular surgeons experienced in BEVAR with both off-the-shelf (e.g., Zenith t-Branch Thoracoabdominal Endovascular Graft by Cook Medical Inc., Bloomington, IN, USA) and custom-made devices (CMDs).

In most cases, a surgical cutdown was performed to expose the common femoral artery (CFA), as well the axillary artery. According to our current standard, lumbar drainage for spinal cord ischemia (SCI) prevention was performed at the surgeon’s discretion [5], depending on the extent of the aortic coverage. For spinal perfusion protection, a mean arterial pressure of >80 mmHg was the aim in accordance with the current European Society for Vascular Surgery (ESVS) guidelines [6]. Additionally, SCI prevention included temporary aneurysm sac perfusion (TASP) [7]. TASP was achieved using a bare metal stent (BMS) as bridging stent, by leaving an open branch, an open iliac limb, or both an open iliac limb and a BMS as a bridging stent. The secondary intervention, which was intended to complete the aneurysm exclusion, was always performed under local anesthesia, with ultrasound-guided percutaneous femoral access and a total femoral approach using a steerable sheath (Aptus HeliFX; Medtronic, MN, USA).

Ballon-expandable Gore Viabahn VBX stent grafts (W.L. Gore & Associates, Flagstaff, AZ, USA) were used as bridging stents in all cases.

Following the intervention, all patients received dual antiplatelet therapy for six months without loading doses. The follow-up protocol included duplex ultrasound examination (DUS), control CTA, and clinical examination at one, six, and twelve months from surgery. After the first year, annually, DUS and CTA were also performed.

Institutional review board approval was requested and obtained. The informed written consent of patients was obtained. Patients gave their consent to publication.

### 2.3. Endpoints

The primary endpoint was technical success, defined as the proper deployment of the stent graft, successful cannulation of the target vessels (TVs), and exclusion of the target pathology without evidence of type 1 or 3 endoleaks, in accordance with the reporting standards for endovascular aortic aneurysm repair [8]. Assisted primary technical success was defined if further unplanned procedures (e.g., due to a type 1A endoleak) were required during the primary procedure to achieve exclusion of the target pathology.

Target vessel success was defined as successful cannulation and bridging stent implantation in the TV without evidence of embolism or dissection and proper branch perfusion [6].

The secondary endpoint was clinical success, defined as the absence of perioperative (≤30 days) major adverse events (MAEs), including acute myocardial infarction (AMI), acute renal injury, respiratory failure requiring invasive or non-invasive ventilation (NIV), acute mesenteric ischemia, stroke, and SCI. Access-site-related complications, as well as ICU- and in-hospital length of stay, were also reported.

The perioperative (≤30 day) and early mortality rates (≤180 day), freedom from aortic re-intervention, TVs patency, and endoleak rates were evaluated during the follow-up, as well as mid-term mortality.

Technical metrics, such as the minimum EIA diameter, iliac tortuosity index (*X*), mean aortic diameter at the E-nside proximal sealing zone, mean aortic diameter at TV ostium level, time for TV catheterization, and the extent of aortic coverage, were retrospectively evaluated.

The iliac artery tortuosity index (*X*) was calculated as *X* = *L*/*D*, where *L* is the centerline length, and *D* is the Euclidean (straight line) distance between its end points [9].

The time for TV catheterization was defined as the duration from the cannulation of the preloaded system (when used) to the deployment of the bridging stent for each vessel.

A descriptive analysis of all variables was performed. SPSS (Version 25; SPSS Inc., Chicago, IL, USA) and Excel (Microsoft Corporation, Redmond, WA, USA) were used for statistical analysis.

A *p*-value < 0.05 was considered statistically significant.

## 3. Results

Between May 2021 and March 2023, 22 patients (16 males, 72.7%) with a mean age of 75.9 ± 5.5 years (range, 69–88) underwent elective iBEVAR for TAAA or PAAA at our University Hospital Center.

The patients’ demographics are listed in Table 1. Even though endovascular repairs often require a more extensive proximal seal than the one required in open surgery, we believe that the clinical outcomes (except for SCI [10]) and patient recovery after surgery are not strictly dependent on the extent of the aortic coverage [11].

The aneurysm extent was Crawford I in four patients (18.2%), III in eight (36.4%), IV in five (22.7%), V in one (4.5%), and PAAA in four (18.2%). All patients presented with degenerative aneurysm. The mean aortic diameter was 63.5 ± 9.9 mm (range, 52–97).

Five patients (22.7%) had a history of prior aortic surgery, including one EVAR, one TEVAR, two open abdominal aortic aneurysm repair, and one frozen elephant trunk.

The technical success rate was 95.5%, with one intraoperative type 1A endoleak observed. A concomitant proximal thoracic endograft was deemed necessary in seven patients. Six patients required TEVAR to extend the proximal sealing due to type I and III TAAA, whereas one patient required TEVAR due to a type 1A endoleak identified at completion angiography. The assisted primary success was 100%.

Eighteen patients (81.8%) received a concomitant bifurcated stent graft, with one patient (4.5%) also receiving an iliac branch device. The internal iliac artery (IIA) was bilaterally patent in nineteen patients; one patient had occlusion of both IIAs, whereas two patients had left IIA occlusion only. Two patients presented with IIA aneurysms, which were successfully embolized with Amplatzer Vascular Plugs-II (Abbott Vascular, CA, USA) without complications.

Seven out of 88 TVs were preoperatively occluded (7.9%), whereas fourteen had a severe stenosis (15.9%). Seventy-eight TVs (96.3%) were successfully incorporated through a bridging stent. One asymptomatic SMA dissection was observed at control CTA (1.3%); one patient died before bridging stent implantation for CT, and another patient experienced CT and left renal artery occlusions before the secondary intervention, due to the improper apposition of the graft fabric over the ostium. The TV success was 95.1%.

A total of 86 bridging stents were used for 78 TVs (1.1 stents/TV). The preloaded system was used for 52 TVs (66.7%). Nine intentional occlusions of inner branches were successfully performed.

### 3.1. Perioperative Outcomes

The clinical success was 86.4%, with three MAEs reported (13.6%). One patient suffered from AMI, one patient suffered from cerebellar ischemia due to distal embolization in the left vertebral artery from the left axillary access, and one developed respiratory failure, requiring NIV and a prolonged ICU stay. Lumbar drainage was never deemed necessary. To prevent SCI, TASP was performed with BMS as the bridging stent in two cases (9.1%), by leaving an open branch in two (9.1%), by leaving an open iliac limb in seven (31.8%), and by leaving both an open iliac limb and BMS as the bridging stent in seven patients (31.8%). TASP was not performed in four patients (18.2%) due to the large size of the aneurysm.

The average time between the primary and secondary intervention was 54.7 ± 30.5 days (range, 13–113). No cases of inter-stage aortic-related mortality or permanent or temporary SCI were reported.

One access-site-related complication occurred due to avulsion of the right EIA during removal of the 24 F introducer sheath, requiring urgent conversion.

The mean ICU length of stay was 1.2 ± 0.8 days (range, 0.25–4), and the mean in-hospital length of stay was 6.6 ± 2.9 days (range, 4–14).

The perioperative and early freedom from all-cause mortality rates were 90.9% and 90%, respectively. Two patients died of heart failure within 30 days from the intervention.

The routine CTA one year after surgery revealed a type III endoleak in two patients. One patient subsequently underwent elective TEVAR, whereas the other declined additional procedures and remains under stricter follow-up. A type II endoleak was detected in three cases, without sac enlargement. The freedom from aortic-related reintervention rate was 95.5%. No bridging stents’ stenosis or occlusion were observed during follow-up, and there was no case of IIA occlusion.

The mean follow-up time was 21.3 ± 12.7 months (range, 9.6–35.3). No cases of aneurysm-related mortality were observed during follow-up. The estimated 1-year survival rate with Kaplan–Meier analysis was 100%, with a TV patency rate of 100% (78/78) (Figure 2).

### 3.2. Technical Metrics

The mean EIA diameter was 7.5 ± 1.1 mm (range, 5.8–10.1), and the mean iliac tortuosity index was 1.34 ± 0.2 (range, 1.1–1.9). Twelve patients (54.5%, including all the female patients) were considered outside the instructions for use (IFU) for the E-nside stent graft due to narrow iliac arteries.

The mean outer-to-outer diameter at the E-nside stent–graft proximal sealing zone, categorized by TAAA class, is listed in Table 2.

The mean aortic diameter at the TV ostium level and the mean time for TV catheterization are listed in Table 3.

The mean operative time was 292.94 ± 61.9 min (range, 155–450); the mean fluoroscopy time was 63.4 ± 15.7 min (range, 39.1–90); the mean contrast volume was 142 ± 40.7 mL (range, 90–200); the mean radiant dose was 30295.9 ± 7386.3 cGy (range, 16,353–43,606).

The mean extent of aortic coverage (above the CT) was 11.7 ± 0.9 cm (range, 10.1–13.2).

## 4. Discussion

The E-nside multibranch stent–graft system is a novel off-the-shelf solution for the treatment of TAAAs and PAAAs, both in elective and urgent settings (Figure 1) [12].

Although CMDs, fenestrated, or polymer-filled-based stent grafts [13] still play a leading role in TAAA and PAAA treatment, this novel off-the-shelf device might fit a wider range of aortic anatomies thanks to its four configurations. A potential downside of off-the-shelf devices (e.g., Cook Zenith T- Branch, E-nside) is that they require a longer sacrifice of healthy aorta and intercostal arteries compared with open repair [14] and CMDs [15], which could lead to a higher rate of permanent or transient SCI.

Nonetheless, because of its configurations, the E-nside stent graft can fix large proximal and distal aorta sealing zones resulting in a sparing of the proximal thoracic aortic coverage and of the intercostal and lumbar arteries’ patency, potentially reducing the risk of SCI. Although the manufacturer’s IFU recommends docking in a thoracic stent graft, we indeed evaluated the feasibility and safety of the E-nside stent graft without proximal thoracic extension.

In our cohort, prior TEVAR was present in two patients (9.1%), whereas a proximal thoracic endograft was deployed in about one third of patients (7/20, 35%). Consistent with the literature [16], our analysis confirms that proximal thoracic endograft deployment is more common in type I-III TAAAs than in type IV-V TAAAs and PAAAs (66.7% vs. 0%, *p* = 0.003).

Notably, a sub-group analysis among patients with type III-IV TAAA and PAAA (*n* = 17) showed that the proximal thoracic stent graft was deployed in four patients with type III TAAA, whereas it was not deemed necessary to extend the proximal sealing in patients with type IV TAAA and PAAA (50% vs. 0%).

In this sub-group of patients, the mean outer-to-outer aortic diameter at the level of the E-nside stent graft’s proximal sealing was 32.2 ± 3.1 mm (range, 26–36; anatomical characteristics of the proximal sealing zone as a function of the aneurysm extent in Table 2). Differently from other off-the-shelf available devices (e.g., Cook Zenith T-Branch), and despite no ancillary proximal fixation (such as barbs), the E-nside stent graft seems capable of also fixing a large proximal thoracic sealing zone (>31 mm), thereby reducing the need for a proximal thoracic stent graft (4/17, 23.5%) and limiting the coverage of the healthy proximal aorta.

The shorter sacrifice of healthy aorta, along with TASP and other adjunctive measures (such as preservation of the antegrade perfusion of the left subclavian artery and IIA and minimization of lower limb ischemia reperfusion injury), may explain the absence of SCI in our study. This rate is significantly lower than the 13.4% reported in a recent meta-analysis of similar devices [17,18].

Piazza M et al. reported an SCI rate of 6.9% with the E-nside stent graft, while Kapalla M. et al. observed complete permanent SCI in 6.8% of patients, immediately after the procedure. In their series, all SCI cases occurred in patients with TAAA (*n* = 2 type II, *n* = 1 type IV), with no cases of transient or late SCI [3,15]. Different from these reports, our series noted no instances of transient or permanent SCI, while the rate of MAEs was consistent with that previously reported (24% vs. 13.6%).

Aortic and iliac axis anatomy remain important limiting factors for the wider applicability of the E-nside stent graft.

Abisi S et al. reported a major suitability of patients with iBEVAR compared to an outer-branched device due to anatomy at the reno-visceral level [19].

In a retrospective analysis evaluating anatomical feasibility, Bilman V et al. reported an overall treatment suitability of 43% with the E-nside stent graft for type I-IV TAAA [20]. The authors concluded that the main limiting factors for its application were the EIA diameter (21%), infrarenal aortic diameter (16%), and the size of the aortic lumen at the level of visceral vessels (14%).

In our series, the mean EIA diameter was 7.5 ± 1.1 mm (range, 5.8–10.1), which is consistently smaller than the manufacturer’s IFU recommendation (≥8.2 mm). Twelve patients (54.5%, including all the female patients) were considered outside the IFU for the E-nside stent graft due to narrow iliac arteries, with eight of these twelve patients presenting an EIA diameter ≤7 mm. Fourteen patients (63.6%) had moderate to severe iliac tortuosity (*X* > 1.25). A narrow iliac axis may lead to suboptimal device orientation, potentially complicating TV catheterization and prolonging the operative time. An extended operative time may increase the arterial wall adhesion to the sheath, further complicating its safe retrieval.

Despite these challenging anatomical features, only one unexpected access-site-related complication was reported (EIA avulsion requiring urgent conversion during the retrieval of the sheath). In this latter case, the EIA accommodated both the insertion and the retrieval of the thoracic stent graft (22 F, requiring an EIA diameter of 8.5 mm), as well as the insertion and the proper deployment of the E-nside stent graft (24 F sheath, 8.2 mm required).

Piazza M. et al. also reported a fatal intraoperative complication due to EIA rupture, which was related to the unsuccessful E-nside deployment caused by difficult advancement of the stent graft through a severely diseased and narrow EIA (<7 mm) [16].

Although the IFU recommends performing a surgical conduit for EIA diameters <8.2 mm, our experience demonstrates the feasibility and safety of E-nside even in the case of a narrow EIA. Nevertheless, alternative strategies are suggested to safely manage the 24 F large bore sheath, especially for female patients. In such cases, we recommend a surgical cutdown to the CFA to enable the retroperitoneal mobilization of the EIA and to straighten the redundant iliac artery. Moreover, multiple serial dilations using sheaths, with a 24 F introducer sheath maintained in place (capable of accommodating off-the-shelf devices), along with continuous retrograde flushing with saline solution, are also critical to minimize the risk of interactions between the devices and the arterial wall. In addition, modifications to delivery systems and sheaths are preferable to address most of the aorto-iliac anatomic limitations and reduce the rate of iliac conduit use and its complications.

Nonetheless, we recommend considering a surgical conduit or iliac endoconduit in cases where the EIA diameter is <7 mm combined with an iliac tortuosity index >1.4 or in the presence of severe circumferential calcifications of the iliac axis, as these conditions may impede the safe insertion and retrieval of the device.

Several reports focused on access-site complications as a major weakness of this stent graft due to the large bore sheath [21,22]. The Inbreed Investigators reported a 90-day access-site reintervention rate of up to 4.3%, whereas Kapalla M et al. reported access-site complications in seven patients (15.9%; including four false aneurysms, three surgical site infections) [3]. In our series, one intraoperative but no late access-site complications were observed. Careful case planning of the access is therefore crucial to improve outcomes and minimize complications. Performing a surgical cutdown to expose the CFA appears to be an effective strategy to reduce the risk of false aneurysms and blood loss in the early postoperative period.

Aortic anatomy at the paravisceral level is the last limiting factor that affects E-nside applicability. In our series, four patients (18.2%) had a lumen diameter <24 mm, consistent with recent reports indicating 15.5% of cases with a narrow aortic diameter <25 mm. No significant differences in terms of TV success and stability were observed between these cases and those with a larger lumen (*p* = 0.23). Precannulation is useful for facilitating the advancement of the sheath into the inner branch and reducing the time required for TV stenting, though no statistically significant differences in catheterization time were noted between cases in which the preloaded system was used and cases in which it was not.

Although Katsargyris A et al. reported that inner branches were often difficult to cannulate [23], our experience aligns with more recent reports indicating that TV success can be easily achieved, from either an axillary or femoral approach, even without a precannulation system.

Our TV success rate was 95.1%, comparable to recently published reports where the TV success rate ranged from 96 to 99%, with an estimated branch patency rate of 98% at twelve months. In our series, no branch-related reintervention was required during the follow-up.

The previous literature on the outcomes of BEVAR has shown a high technical success rate and a low incidence of perioperative morbidity and mortality [15,24,25]. To date, there are few available reports in the literature on the early- to mid-term results of the E-nside stent graft [3,16,18,26,27,28].

Yazar O et al. [27] reported a series of 23 patients treated with the E-nside stent graft over a period of 56 months and a mean follow-up time of 15 months with a technical success of 96%. Our data align with these findings, with a technical success rate of 95.5% and an assisted primary technical success rate of 100%.

Our perioperative mortality rate was 9.1% (2/22), slightly higher than that reported by Yazar O [27] (8.3%) and by Piazza M [16] (5.2%). Nevertheless, our cohort was older (75.9 vs. 73 years), with seven patients (mean age of 80.3 years) classified as frail [1], having histories of stroke and moderate-to-severe cognitive impairment. As reported, frail patients have a worse ability to recover after surgery, especially after MAE or procedure-related complications [1]. Our frail patients experienced significantly longer hospitalization and were more frequently non-home discharged compared to the fit patients. This increase in mortality may not only reflect the complications following iBEVAR but also the greater risks inherent to longer hospitalization (such as immobilization, loss of muscle mass, and functional and cognitive decline) that are strictly related to frailty.

### Limitations

This research has certain limitations. First, it is subject to biases related to a retrospective data collection, as well as limitations related to the small sample size and the retrospective non-randomized single-center study design. Additionally, over the two-year study period, a learning curve and an increase in the proficiency with this endovascular technique and device may have influenced both the technical and clinical outcomes.

## 5. Conclusions

Our initial experience confirms that the E-nside thoraco-abdominal multibranch stent–graft system shows promising early technical and clinical success rates, along with low reintervention rate.

However, drawbacks include the extensive sacrifice of the healthy aorta and the 24 F delivery system, which represent a limitation in patients with narrow iliac arteries. Nonetheless, our analysis demonstrated a low rate of access-site complications, even in patients treated outside the IFU recommendations for iliac anatomy.

The preservation of the precannulation system, as well as the reduction in the diameter of the delivery system, are crucial to establish the E-nside as a leading endograft for complex aortic disease, particularly in female patients.

Mid- to long-term confirmatory larger studies and prospective registries are mandatory to assess the long-term efficacy and safety of the E-nside stent graft, as well as to compare it with other off-the-shelf solutions, such as in situ fenestration, chimney techniques, or outer-branched stent grafts.

## Figures and Tables

**Figure 1 diagnostics-14-02612-f001:**
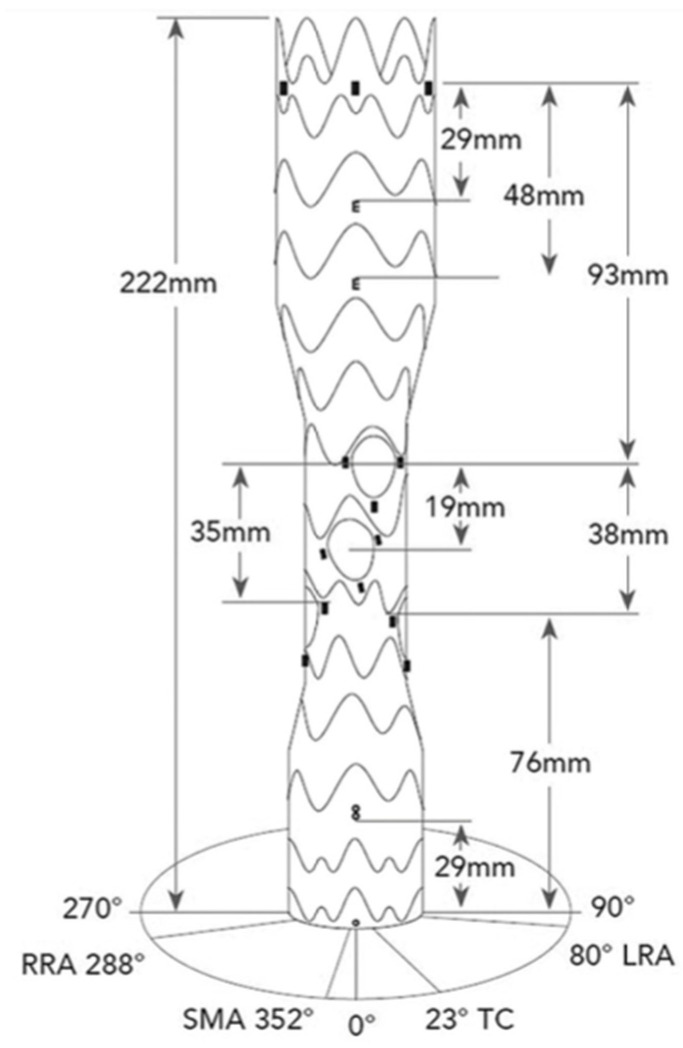
Illustration of the different markers, distances, and orientation of the outlets of the inner branches of the E-nside multibranch stent–graft system. CT, celiac trunk; SMA, superior mesenteric artery; RRA, right renal artery; LRA, left renal artery. The prosthesis is available in four different configurations with proximal diameters of 38 and 33 mm and distal diameters of 30 and 26 mm. All rights reserved. For permission: Artivion Inc., Kennesaw, GA, USA.

**Figure 2 diagnostics-14-02612-f002:**
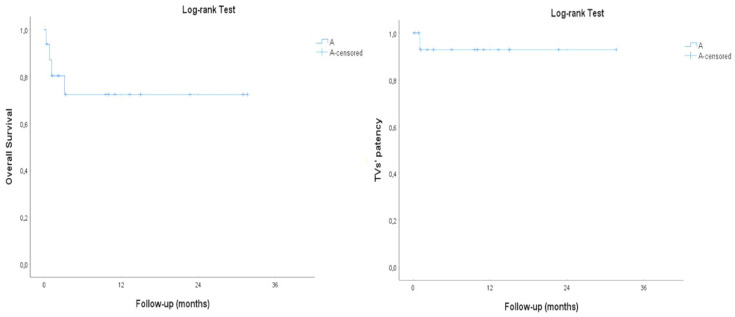
Survival curve with Kaplan–Meier analysis. The estimated 1-year survival rate was 100%, with a target vessel (TV) patency rate of 100% at one year.

**Table 1 diagnostics-14-02612-t001:** Patients’ demographics.

Age-y (Mean, SD, Range)	75.9 ± 5.5 (69–88)
Male (*n*, %)	16 (72.7)
Arterial hypertension (*n*, %)	22 (100)
Diabetes (*n*, %)	3 (13.6)
Dyslipidemia (*n*, %)	21 (95.4)
Smoking habit (*n*, %)	15 (68.2)
Coronary artery disease (*n*, %)	12 (54.5)
Chronic renal failure (*n*, %)	10 (45.4)
COPD (*n*, %)	12 (54.5)
Malignancy (*n*, %)	6 (27.3)

**Table 2 diagnostics-14-02612-t002:** Outer-to-outer diameter at the E-nside stent-graft proximal sealing zone, categorized by aneurysm extension.

	Type I TAAA *n* = 4	Type II TAAA *n* = 0	Type III TAAA *n* = 8	Type IV TAAA *n* = 5	Type V TAAA *n* = 1	PAAA *n* = 4
Outer-to-outer diameter (mm) at the level of proximal sealing	/	/	33.2 ± 3.8	31 ± 2.5	/	31.5 ± 0.7

**Table 3 diagnostics-14-02612-t003:** Mean aortic diameter at target vessel ostium level and mean time for target vessel catheterization. CT = celiac trunk; SMA = superior mesenteric artery; RRA = right renal artery; LRA = left renal artery.

	Mean Aortic Diameter at Target Vessel Ostium Level (mm, Range)	Mean Time of Target Vessel Catheterization (min, Range)
CT	30.4 (24.5–44.2)	20.7 (10–60)
SMA	28.7 (21.3–40.4)	15.5 (7–24)
RRA	29.4 (19.7–54)	21.1 (8–39)
LRA	31.0 (19–54)	21.1 (10–41)

## Data Availability

The data presented in this study are available on request from the corresponding author. The data are not available due to privacy restrictions.

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
