# Peer review of "Early Experience with Inner Branch Stent–Graft System for Endovascular Repair of Thoraco-Abdominal and Pararenal Abdominal Aortic Aneurysm"

_diagnostics, 2024, doi:10.3390/diagnostics14232612_

Round 1
Reviewer 1 Report
Comments and Suggestions for Authors
This is a well conducted review of a single institutional experience with the E-nside aortic stent graft. The analysis was well conducted but only contained elective cases. Have you also performed more urgent cases and do you you alter your adherence o IFUs for urgent cases? For elective cases the early mortality of 10% seems slightly high. Do you think that precannulation is necessary or helpful since you state you have no difficulty with cannulation,?
Author Response
Comment:
This is a well conducted review of a single institutional experience with the E-nside aortic stent graft. The analysis was well conducted but only contained elective cases. Have you also performed more urgent cases, and do you you alter your adherence or IFUs for urgent cases? For elective cases the early mortality of 10% seems slightly high. Do you think that precannulation is necessary or helpful since you state you have no difficulty with cannulation?
Response:
In our experience, we have performed two procedures for ruptured TAAA/PAAA with Enside stent-graft, that we excluded from this report.
For urgent/emergent cases, we are less strictly adherent to the IFUs. In these cases, we account the Enside feasibility most for TVs’ catheterization (anatomical configurations/orientations of TVs) than for EIA diameter or aortic diameter at renovisceral level. In these latter cases, multiple serial dilation with dryseal sheaths and partial deployment of the stent-graft represent the adjunctive procedure to adress these issues.
10% of early mortality rate is high. Nevertheless, as discussed, our patients were older than what reported in Literature (75.9 vs 73 years). Moreover, seven patients (31.8%) were older than 80 years old and frail. As known, moderate to severe frailty negative affect the outcomes, especially in term of early mortality.
Precannulation is very helpful to facilitate TVs catheterization, but not always necessary. In cases of narrow EIAs, we prefer to remove the delivery system within the artery to reduce the risks of SCI and acute limb ischemia. Thus, in these cases we proceeded with anterograde cannulation of directional branches and TVs without precannulation. No statistically significance emerged in term of TVs success and in time of TVs catheterization.
Reviewer 2 Report
Comments and Suggestions for Authors
I have received for review an original research article entitled “Early Experience with Inner Branch Stent-Graft System for Endovascular Repair of Thoraco-Abdominal and Pararenal Abdominal Aortic Aneurysm” prepared by Simone Cuozzo et al., which had been submitted to Diagnostics (IF=3.0). Cardiovascular diseases are one of the most important challenges for public health, as they are the leading cause of morbidity and mortality in many countries around the world. The most important problems include diseases in the course of atherosclerosis. Although the "atherosclerotic etiology" of aortic aneurysms is less and less often discussed, a large part of aortic aneurysms develops as a result of cardiovascular risk factors, which also lead to the development of atherosclerosis, such as arterial hypertension, dyslipidemia, diabetes, smoking, low physical activity. The topic taken up by the Authors of the manuscript is therefore extremely important. Patients with aortic aneurysms who qualify for surgical treatment are usually patients with a very poor general condition of the cardiovascular system, which significantly worsens the prognosis, especially in the case of surgical treatment, hence endovascular techniques are becoming increasingly important. The development of new techniques for the endovascular treatment of aortic aneurysms is therefore of great importance. In my opinion, the presented manuscript is well prepared and presents high substantive and cognitive value. I only think that the text should undergo linguistic correction. I have no substantive objections.
Comments on the Quality of English LanguageSome English corrections are necessary. For instance, in the line 45 there is “as well know” whereas in my opinion it should be “as well known”.
Author Response
Comments:
I have received for review an original research article entitled “Early Experience with Inner Branch Stent-Graft System for Endovascular Repair of Thoraco-Abdominal and Pararenal Abdominal Aortic Aneurysm” prepared by Simone Cuozzo et al., which had been submitted to Diagnostics (IF=3.0). Cardiovascular diseases are one of the most important challenges for public health, as they are the leading cause of morbidity and mortality in many countries around the world. The most important problems include diseases in the course of atherosclerosis. Although the "atherosclerotic etiology" of aortic aneurysms is less and less often discussed, a large part of aortic aneurysms develops as a result of cardiovascular risk factors, which also lead to the development of atherosclerosis, such as arterial hypertension, dyslipidemia, diabetes, smoking, low physical activity. The topic taken up by the Authors of the manuscript is therefore extremely important. Patients with aortic aneurysms who qualify for surgical treatment are usually patients with a very poor general condition of the cardiovascular system, which significantly worsens the prognosis, especially in the case of surgical treatment, hence endovascular techniques are becoming increasingly important. The development of new techniques for the endovascular treatment of aortic aneurysms is therefore of great importance. In my opinion, the presented manuscript is well prepared and presents high substantive and cognitive value. I only think that the text should undergo linguistic correction. I have no substantive objections.
1.Some English corrections are necessary. For instance, in the line 45 there is “as well know” whereas in my opinion it should be “as well known”.
Response:
Thank you.
Reviewer 3 Report
Comments and Suggestions for Authors
Dear Authors, congratulations for reporting your early experience with a novel branched endovascular device.
This is a descriptive study reporting on technical aspects involved, patient population targeted, and outcomes observed during a 2-year practice of thoracoabdominal aneurysm repair with the E-nside device. The findings of this study has potential value for being used as reference by future adopters of the device, and also there is potential for being included in future meta analyses. While the manuscript is to be credited for its clear reporting of the technicalities and detailed presentation of outcomes, there are some opportunities for improvement. Please see these as follows:
1. Line 26: "Twelve patients (six women, 100%)": I guess this means 6/6 of women? Please clarify on that. The statement is difficult to understand in its current form, as it forces the reader to think 6/12 is %100!
2. Line 68: Please explain abbreviation the first time they are used (e.g., spinal chord ischemia [SCI]).
3. Lines 66-68: Even though endovascular repairs often require a more ex- 66 tensive proximal seal than the one required in open surgery, we believe that the clinical 67 outcomes (except for SCI [1]) and patient recovery after surgery are not strictly dependent 68 of the extent of the aortic coverage [2]. -- This sentence does not belong in the Methods section.
4. Figure 1: Is this figure original, or excerpted from another source? I the latter is true, please obtain permission from the publisher and state it in the figure caption.
5. Line 115: This is a retrospective study. How come you obtained informed consent from the participants with regards to publication of this manuscript? Please clarify on this.
6. Line 145: I do not think that there is any need for a P value anywhere within this manuscript. This sentence should be omitted.
7. Line 147: Report either males or females, not both.
8. Line 179: (n=X, %). Needs to be corrected.
9. Lines 249-250: Proximal stent-graft was deployed in 4/8 (%50) of type III aneurysm patients, and 0/5 (%0) of type IV aneurysm patients. Providing a P value seems inappropriate when comparing events with so few occurrences. Please reconsider omitting the P value.
10. Line 316: (p=ns) -- explain this.
11. Line 332: Yazar et al -- needs citation number.
12. Line 336: Yazar et al and Piazza et al -- need citation numbers.
Thank you for submitting your paper to Diagnostics. I am glad to be involved in the review process. I look forward to reviewing a revised version.
Comments on the Quality of English Language
I strongly recommend proof-reading by a native speaker who has experience in medical publishing.
Author Response
This is a descriptive study reporting on technical aspects involved, patient population targeted, and outcomes observed during a 2-year practice of thoracoabdominal aneurysm repair with the E-nside device. The findings of this study has potential value for being used as reference by future adopters of the device, and also there is potential for being included in future meta analyses. While the manuscript is to be credited for its clear reporting of the technicalities and detailed presentation of outcomes, there are some opportunities for improvement. Please see these as follows:
1. Line 26: "Twelve patients (six women, 100%)": I guess this means 6/6 of women? Please clarify on that. The statement is difficult to understand in its current form, as it forces the reader to think 6/12 is %100!
That’s right. Thank you for your comment.
“Twelve patients (54.5%, including all female patients) were considered outside the instructions for use (IFU) due to narrow iliac arteries.”
2. Line 68: Please explain abbreviation the first time they are used (e.g., spinal cord ischemia [SCI]).
Done, Thank you.
3. Lines 66-68: Even though endovascular repairs often require a more ex- 66 tensive proximal seal than the one required in open surgery, we believe that the clinical 67 outcomes (except for SCI [1]) and patient recovery after surgery are not strictly dependent 68 of the extent of the aortic coverage [2]. -- This sentence does not belong in the Methods section.
Ok. Done.
4. Figure 1: Is this figure original, or excerpted from another source? I the latter is true, please obtain permission from the publisher and state it in the figure caption.
Done, Thank you. I’ve added the information in the text. All rights reserved. For permission: Artivion Inc., Kennesaw, GA, USA.
5. Line 115: This is a retrospective study. How come you obtained informed consent from the participants with regards to publication of this manuscript? Please clarify on this.
Some Journals need written informed consent of patients also for retrospective analysis. I’ve informed the patients regarding the study and added this information during follow-up clinical examination.
6. Line 145: I do not think that there is any need for a P value anywhere within this manuscript. This sentence should be omitted.
Ok, thank you. I’ve added a p-value (0.23) in the text. If you don’t need, I’ll remove this sentence.
7. Line 147: Report either males or females, not both.
Done.
8. Line 179: (n=X, %). Needs to be corrected.
N=3, 13.6%; Thank you.
9. Lines 249-250: Proximal stent-graft was deployed in 4/8 (%50) of type III aneurysm patients, and 0/5 (%0) of type IV aneurysm patients. Providing a P value seems inappropriate when comparing events with so few occurrences. Please reconsider omitting the P value.
Ok, I’ve removed the p-value.
10. Line 316: (p=ns) -- explain this.
p-value about this data is 0.23. I’ve added the value and removed ns=not significant.
11. Line 332: Yazar et al -- needs citation number.
Thank you, added.
12. Line 336: Yazar et al and Piazza et al -- need citation numbers.
Thank you, added.